# Therapies for the Treatment of Advanced/Metastatic Estrogen Receptor-Positive Breast Cancer: Current Situation and Future Directions

**DOI:** 10.3390/cancers16030552

**Published:** 2024-01-27

**Authors:** Rohan Kalyan Rej, Joyeeta Roy, Srinivasa Rao Allu

**Affiliations:** 1Rogel Cancer Center, Department of Internal Medicine, University of Michigan, Ann Arbor, MI 48109, USA; srallu@med.umich.edu; 2Departments of Medicinal Chemistry, College of Pharmacy, Rogel Cancer Center, University of Michigan, Ann Arbor, MI 48109, USA; joyeetar@umich.edu

**Keywords:** breast cancer, estrogen receptor, endocrine therapy, oral SERDs, CDK4/6 inhibitors

## Abstract

**Simple Summary:**

The estrogen receptor (ER) plays a critical role in the initiation and progression of breast cancer. Utilizing specialized therapies aimed at the ER has been effective in many instances and is commonly employed in breast cancer treatment protocols. The selection of therapy depends on multiple factors, including the menopausal status, breast cancer stage, and unique tumor attributes. These therapies can function independently as monotherapy, or in conjunction or sequential alignment with other treatments, based on the distinct characteristics of the breast cancer and the patient’s overall health. Furthermore, current research endeavors are focused on producing newer and more precise therapies for breast cancer, which include strategies to combat resistance mechanisms. The objective of this review is to provide a snapshot of the existing landscape and sketch out future paths for the progression of HR+ breast cancer treatments currently under clinical development.

**Abstract:**

The hormone receptor-positive (HR+) type is the most frequently identified subtype of breast cancer. HR+ breast cancer has a more positive prognosis when compared to other subtypes, such as human epidermal growth factor protein 2-positive disorder and triple-negative disease. The advancement in treatment outcomes for advanced HR+ breast cancer has been considerably elevated due to the discovery of cyclin-dependent kinase 4/6 inhibitors and their combination effects with endocrine therapy. However, despite the considerable effectiveness of tamoxifen, a selective estrogen receptor modulator (SERMs), and aromatase inhibitors (AI), the issue of treatment resistance still presents a significant challenge for HR+ breast cancer. As a result, there is a focus on exploring new therapeutic strategies such as targeted protein degradation and covalent inhibition for targeting ERα. This article discusses the latest progress in treatments like oral selective ER degraders (SERDs), complete estrogen receptor antagonists (CERANs), selective estrogen receptor covalent antagonists (SERCAs), proteolysis targeting chimera (PROTAC) degraders, and combinations of CDK4/6 inhibitors with endocrine therapy. The focus is specifically on those compounds that have transitioned into phases of clinical development.

## 1. Introduction

Breast cancer continues to be the most prevalent type of cancer among women, constituting to roughly 25% of all newly diagnosed cases and 16% of global cancer-related deaths. Regrettably, advanced breast cancer that has metastasized to distant organs is beyond a cure with the treatments currently available. According to the American Cancer Society, projections for the US in 2023 indicate that approximately 297,790 women will be diagnosed with invasive breast cancer and about 43,700 will succumb to the disease. Advancements in the treatment of breast cancer have decreased death rates by 43% from 1989 to 2020, translating to 460,000 less breast cancer deaths during that time [1]. The surrogate intrinsic subtypes are typically used clinically and are based on histology and immunohistochemistry expression; the subtypes of breast cancer have been identified as Luminal A-like, Luminal B-like HER2−, Luminal B-like HER2+, HER2-enriched, and Triple-negative. Luminal A-like accounts for 60–70% of breast cancer cases in the United States, followed by Luminal B-like HER2− at 10–20%, triple negative at 10–15%, and HER2-enriched and Luminal B-like HER2+ together at approximately 13–15% (Figure 1) [2,3]. Among the different types, Luminal A tumors, the most prevalent molecular subtype, tend to grow at a slower pace compared to the other cancer types. Luminal A breast cancers are a subtype of breast cancer that are typically hormone-receptor-positive (ER/PR-positive), HER2-negative, low proliferation and a low-risk GES (gene expression signature). This means they generally respond well to endocrine therapy (ET) and have a better prognosis compared to other subtypes [4]. By contrast, the Luminal B-like group is HR-positive but have an ER and PR expression lower than Luminal A-like, but it exhibits high grade or high proliferation and high-risk GESs. Whereas TNBC and non-luminal types demonstrate aggressive characteristics such as high grade, lack of ER, PR and HER2 expression, and high proliferation, making them a distinct breast cancer subtype tied to poorer prognoses [5]. Despite numerous attempts to find potential therapeutic targets, chemotherapy continues to be the primary systemic treatment for TNBC. The development of specialized agents for TNBC has not progressed as much as those targeting ER and HER2 in other clinical subtypes. However, substantial recent progress has been made with innovative treatments, such as poly adenosine diphosphate (ADP)-ribose polymerase inhibitors (PARPi) for patients with germline BRCA1/2 mutations. These inhibitors were approved by the FDA in 2019 for metastatic disease and later in 2021 for early stages of the disease [6,7,8]. The first immunotherapy regimens for TNBC were also approved for metastatic disease beginning in 2019 [9] and for early disease in 2021 [10]. The HER2-enriched (HR−/HER2+) subtype was associated with the worst prognosis in the past. However, the use of targeted therapies for HER2+ cancers has significantly improved outcomes for these patients. Trastuzumab, which was introduced 25 years ago, has become the standard of care treatment and has greatly improved the treatment of HER2+ breast cancer patients [11]. Trastuzumab functions by attaching to the IV domain of the HER2 receptor, which blocks downstream signaling and triggers both antibody-dependent cell-mediated cytotoxicity and antibody-dependent cell-phagocytosis. Presently, Trastuzumab is given alongside chemotherapy or radiotherapy for one year in an adjuvant setting. In a neo-adjuvant context, Trastuzumab is combined with Pertuzumab, a monoclonal antibody that aims at the II domain of the HER2 molecule, suppressing its ability to dimerize [12]. However, despite Trastuzumab showing improved responses and outcomes, a significant portion of patients still develop resistance to the therapy and experience a recurrence of the disease [13]. Given these clinical results, considerable endeavors have been made to develop new therapies targeting HER2. These include monoclonal antibodies targeting different HER2 epitopes, antibody-drug conjugates, bispecific antibodies, tyrosine kinase inhibitors (TKIs), and others [14]. Efforts are also ongoing to stimulate the immune response in HER2+ patients, following the evident benefits derived from immunotherapy in TNBC. Numerous strategies are being utilized to achieve this goal, such as the administration of checkpoint inhibitors, the connection of effector T cells with HER2 antibodies, the implementation of cellular therapy, and the use of vaccines [3,15]. As for HR+ breast cancer treatment, endocrine therapy remains a cornerstone, but improvements in patient outcomes have been achieved by introducing novel therapeutic agents that act on extra-hormonal molecular targets—such as cyclin-dependent kinase 4/6 (CDK4/6), the mammalian target of rapamycin (mTOR), protein kinase B (also known as Akt), and phosphatidylinositol 3-kinase (PI3K). However, advanced HR+ breast cancer still deems uncurable. It often manifests through mechanisms that are independent of hormonal signaling, leading to recurrence, treatment resistance, and potential metastasis, thus accentuating the necessity for additional translational research. This review will focus on the existing and emerging therapeutic agents for advanced HR+ breast cancer, emphasizing the key studies that have facilitated their integration into clinical practice.

## 2. Endocrine Therapy: Treatment for ER+/HER2− MBC

Endocrine therapy is typically categorized into three classes. The first class of drugs used are SERMs, with tamoxifen serving as a prominent example. SERMs act as competitive inhibitors of estrogen-ER binding and display either agonist or antagonist activity, depending on the target tissue. These drugs function by inhibiting the binding of estrogens and preventing ER signaling. While SERMs are effective at blocking the activation function AF2 domain, they are incapable of blocking the AF1 domain of ERα, which can lead to agonist activity and limit its effectiveness [17,18]. The second class comprises the 3rd generation aromatase inhibitors (AIs), which function by inhibiting estradiol biosynthesis, thereby preventing ER signaling. Nonsteroidal AIs, such as anastrozole and letrozole, function as noncovalent, reversible competitive inhibitors of androgen–aromatase binding. Conversely, the steroidal AI, such as exemestane, covalently and irreversibly binds to the aromatase substrate-binding site [19]. The third class is the SERDs, of which fulvestrant and elacestrant are the current approved drugs in this category [20,21]. SERDs are high-affinity competitive antagonists that block ER dimerization and DNA binding, inhibit nuclear uptake, and increase the turnover and degradation of the ER. This overall leads to an improved inhibition of estrogen signaling. CDK4/6 inhibitors are often used in conjunction with AIs or SERDs to enhance efficacy, particularly for advanced or metastatic breast cancer (MBC). To enhance efficacy, especially for advanced or metastatic breast cancer, CDK4/6 inhibitors are often used in tandem with AIs or SERDs. It has been evidenced that combining CDK4/6 inhibitors with endocrine therapy boosts the objective response rate (ORR), progression-free survival (PFS), and overall survival (OS) in ER positive MBC. This improvement was observed when CDK4/6 inhibitors were added to an AI for first-line ET or fulvestrant as a second-line ET, after progression or relapse on an AI [22,23,24]. The approval of fulvestrant and elacestrant represent significant advancements as pure ERα antagonist drugs and led the discovery of a new class of anti-estrogen drugs, including CERANs, SERCAs, and PROTACs that target ER (Figure 2, Table 1). CERANs work by impeding all agonist signaling facilitated by ERα through the total inactivation of the ER by suppressing both activation functions of ER transcription (AF-1 and AF-2). As a result, they antagonize the ER, block transcriptional activity, and promote ER degradation [25]. On the other hand, SERCAs contain a Michael acceptor (e.g., arylamide chain) to attach covalently to a cysteine residue (C530) of the ER, thereby functioning as covalent inhibitors of ER transcription (Figure 2) [26]. PROTACs function by binding both an E3 ubiquitin ligase and a target protein concurrently; thus, bringing the ubiquitin ligase complex into proximity to the target protein. This proximity enables the transfer of ubiquitin from E2 to the target protein, thus triggering its eventual degradation by the proteasome (Figure 2). In recent years these innovative approaches have emerged as a leading modality in drug discovery [27,28,29].

## 3. Prevalence of *ESR1* Mutations in Metastatic Breast Cancer

Endocrine therapy is typically successful in treating the majority of patients with hormone receptor-positive (HR+) advanced breast cancer. However, over time, cancer cells may develop resistance to this therapy, either by acquiring new mutations or losing hormone receptor expression [30]. Primary endocrine resistance is typically characterized by a relapse within the first two years of adjuvant endocrine therapy or disease progression within the initial six months of endocrine therapy for advanced or metastatic breast cancer. On the other hand, secondary or acquired resistance is identified as a relapse that transpires after a minimum of two years of endocrine therapy, either during or within the first year of completing adjuvant endocrine therapy [4,31].

*ESR1* mutations represent one of the most well-known and extensively investigated molecular mechanisms of therapeutic resistance. These mutations can cause ligand-independent activity, fostering tumor growth and resistance to ET. The prevalence of such mutations depends on the duration of ET and can be found in 20%–40% of patients previously treated with aromatase inhibitors (AIs) for MBC. However, these mutation rates are much lower in the case of recurrent breast cancer and less than 1% in ET-naive patients. This suggests that *ESR1* mutations may be acquired during AI treatment in the metastatic setting [32]. In the PADA-1 trial, it was observed that patients with pre-existing *ESR1* mutations undergoing AI and CDK4/6 inhibitor therapy for MBC experienced an increase in the *ESR1* mutation, reaching up to 27% at a median time of 15.6 months (Table 2) [33]. In the BOLERO-2 clinical trial, which enrolled 541 patients, 29% of participants (156 patients) exhibited a mutation in the estrogen receptor. Interestingly, patients with the mutation had a significantly shorter median OS (20.7 months) than those without the mutation (32.1 months) [34]. Other established ET resistance mechanisms include the growth-promoting PI3K-AKT-mTOR and RAS/RAF/MEK/ERK pathways [35]. To improve the prediction of resistance, there is a need for more extensive, real-time monitoring of the dynamic mutational patterns, both during and after treatment. This necessitates employing methods that involve examining additional prognostic biomarkers like *ESR1*, PIK3CA, and AKT mutations. Real-time monitoring can be implemented through diverse techniques, such as liquid biopsy methods that analyze circulating tumor DNA (ctDNA) or other liquid biopsy markers. These strategies offer a non-invasive option and potentially enable more frequent evaluations of the tumor’s genetic makeup, allowing for timely adjustments to the treatment plan in response to evolving genomic alterations.

## 4. Data from Early Clinical Trials and Lessons Learned

Multiple early-phase clinical trials have explored oral SERDs featuring acrylic acid side chains, including GDC-0810, AZD9496 and LSZ102 (Table 1) [38]. These agents have demonstrated significant antitumor activity in both endocrine-sensitive and resistant preclinical models, as well as in *ESR1*-mutated tumors [39,40,41,42,43]. Despite their potential, early-phase clinical trials have unveiled challenges indicating lower efficacy and tolerability, leading to the limited progression of most of these compounds beyond phase I.

For example, the clinical trial (NCT02569801) of GDC-0810 was terminated due to its inferior effectiveness compared to fulvestrant in a phase II study. Despite its ability to degrade ERα significantly, reaching 91%, GDC-0810 lacks a complete antagonist profile and weakly activates ER transcription, potentially impacting in vivo efficacy [44]. The clinical trials for AZD9496 have shown low response rates and an unfavorable toxicity profile. Considering the modest clinical benefits and unfavorable toxicity profile, the development of AZD9496 was halted in February 2021, opting instead for its more potent and well-tolerated successor, camizestrant [45]. Likewise, the phase I clinical trial of LSZ102 (NCT02734615) was discontinued due to interim single-agent results, indicating an ORR of only 1.3% in heavily pretreated ER+ metastatic breast cancer patients. However, when LSZ102 was used in combination with ribociclib, a CDK4/6 inhibitor, alpelisib, or a PI3K inhibitor, the overall response rate improved to 17% or 7%, respectively [46,47].

Several additional SERDs demonstrating preclinical activity are under evaluation in phase I–II clinical trials, including taragarestrant (NCT03471663) [48], ZN-c5 (NCT03560531) [49], GDC-0927 (NCT02316509) [50], SCO-120 (NCT04942054) [48]. But unfortunately, the degradative properties did not translate into a therapeutic benefit, leading to its discontinuation or halt in development; this highlights the challenges in creating new molecular entities that encompass the full set of desirable features.

## 5. The Rise of Oral SERDs: Data from New Generation Novel SERD

For nearly two decades, fulvestrant stood as the sole approved SERD for the treatment of metastatic breast cancer (MBC) before the FDA approval of elacestrant. It achieves its antagonism of ERα through receptor downregulation and degradation. Fulvestrant has been effective in treating tumors that have progressed on previous endocrine therapies, such as tamoxifen or AIs, suggesting that its degradation of ERα leads to better outcomes than simple antagonism [23,51]. However, there are limitations to the use of intramuscularly administered fulvestrant due to its suboptimal pharmacokinetic properties. Even with a higher dose, about 50% of the ERα protein remains when the tumor biopsies were compared at baseline and after 4 weeks of treatment with the higher dose of fulvestrant [52]. A set of new generation novel SERDs has been developed to address these challenges, boasting key advantages such as enhanced potency, oral formulations, efficacy in post-endocrine therapy (ET) and post-CDK4/6 inhibitor (CDK4/6i) disease, and activity in *ESR1*-mutant MBC.

Borestrant, also known as fulvestrant-3-boronic acid, was derived from fulvestrant by converting the phenolic hydroxyl in fulvestrant to a boronic acid, enhancing its oral bioavailability and clinical efficacy. The ongoing first-in-human study ENZENO (NCT04669587), is assessing the safety and tolerability of Borestrant, both as a single agent and in combination with palbociclib, in patients with ER+ MBC [53]. Rintodestrant represents one of the few acrylic acids side chains SERD currently evaluated in phase I as a monotherapy and combined with Palbociclib (NCT03455270) [54].

Elacestrant, amcenestrant, camizestrant, giredestrant and imlunestrant, which employ a basic amine group to replace the acid group, exhibit enhanced tolerability and have fewer side effects, making them advanced oral SERD molecules. In January 2023, the FDA approved elacestrant for the treatment of postmenopausal women or adult men. This approval was specifically for individuals with ER+, HER2−, and *ESR1*-mutated advanced or metastatic breast cancer, who exhibited disease progression following at least one course of endocrine therapy. This approval was endorsed by the phase III EMERALD trial (NCT03778931), which was the first study to demonstrate greater efficacy for an oral SERD over fulvestrant. The trial has assessed elacestrant compared to SOC (fulvestrant or aromatase inhibitor) in patients who had progressed on both a CDK4/6 inhibitor and an endocrine therapy. Elacestrant exhibited an extension of 12 month PFS and a median PFS in the overall population (22.3% vs. 9.4%; 2.8 vs. 1.9 months) and an *ESR1*-mutation cohort (26.8% vs. 8.2%; 3.8 vs. 1.9 months) fulfilling the primary endpoints of the study. This study suggested that patients with *ESR1* mutations may derive greater benefits from elacestrant compared to other endocrine monotherapies. These findings are expected to influence the SOC in the second-line metastatic setting, endorsing a precision medicine approach, particularly when other targetable mutations are present [55]. Results from a phase I study (NCT02338349) revealed that elacestrant exhibits single-agent activity in heavily pretreated patients with ER+ MBC [56]. Furthermore, ongoing investigations include the assessment of elacestrant in combination with abemaciclib in patients with brain metastases (NCT04791384) and assessing alterations in Ki67 within the presurgical setting (NCT04791384) [57,58].

However, there were also some notable setbacks for these new generation oral SERDs. For instance, in August 2022, Sanofi discontinued the clinical development of amcenestrant based upon the interim analysis of the phase III AMEERA-5 trial (NCT04478266) evaluating amcenestrant in combination with palbociclib vs. letrozole combined with palbociclib in patients with estrogen receptor-positive ER+/HER2− advanced breast cancer. The randomized, double-blind study involved 1068 patients, with outcomes showing discontinuation of amcenestrant due to efficacy concerns. Five months earlier, in March 2022, amcenestrant failed in the phase II AMEERA-3 trial (NCT04059484), as it did not meet the primary endpoint of improving PFS compared to the physician’s chosen endocrine treatment. This trial focused on patients with locally advanced or metastatic ER+ and HER2− breast cancer who had progressed after hormonal therapies. Additionally, other studies of amcenestrant, such as AMEERA-6 (NCT05128773) which is evaluating it in early-stage breast cancer, will be discontinued. The randomized, double-blind AMEERA-6 trial compared amcenestrant to tamoxifen in patients with hormone receptor-positive, early breast cancer who discontinued adjuvant AI therapy due to toxicity (Table 3) [48,59].

Additionally, the acelERA randomized phase II trial (NCT04576455), which compared giredestrant to a physician’s choice of endocrine therapy, failed to show a significant PFS benefit in a similar population when compared to the EMERALD and AMEERA-3 clinical trial [60]. Despite this, Roche highlighted that the efficacy data for giredestrant was encouraging, particularly in patients with a higher dependence on estrogen receptor activity.

Nevertheless, some encouraging clinical data have emerged. Results from coopERA phase II window-of opportunity trial (NCT04436744) for amcenestrant indicate superior response to giredestrant vs. anastrozole for patients with early ER+/HER2− BC. Interim analysis results (of 83/202 patients) showed a more significant relative reduction in Ki67 at two weeks (80% vs. 67%) and tumors achieving a complete cell cycle arrest (25% vs. 5.1%) with giredestrant vs. anastrozole. These results suggest that giredestrant was better at suppressing tumor cell proliferation compared to anastrozole. Notably, this trial represents the first comparison between an aromatase inhibitor and an oral SERD, meeting its primary endpoint [61].

Building on this proof-of-concept data, a phase III trial is currently exploring giredestrant in both the adjuvant setting (lidERA; NCT04961996) and the metastatic setting (Persevera; NCT04546009), aiming to compare its efficacy with standard endocrine therapy [62]. The ongoing phase I study (NCT03332797) revealed that giredestrant, when used alone and in combination with palbociclib, has demonstrated promising activity in advanced/metastatic breast cancer. As a monotherapy, giredestrant displayed an ORR of 13% and a median PFS of 7.8 months. When combined with palbociclib, the ORR increased to 33%, with a median PFS of 9.3 months. Adverse events (AEs), primarily grades 1–2, included fatigue, arthralgia, nausea, bradycardia, and occasional visual impairment (Table 3) [63].

Camizestrant is currently under investigation in the phase I SERENA-1 study (NCT03616587) in woman with advanced breast cancer. The initial findings from 98 patients, who had undergone prior treatment with fulvestrant and CDK4/6 inhibitors, revealed an ORR of 10%, a CBR of 35%, and a median PFS of 5.4 months. Within the cohort of patients naive to CDK4/6 inhibitors, which comprised 25 individuals receiving a combination of camizestrant and palbociclib, the ORR and CBR stood at 5.9% and 28%, respectively [64,65].

The SERENA-2 study (NCT04214288) evaluated camizestrant against fulvestrant, while the SERENA-4 and SERENA-6 studies (NCT04711252; NCT04214288) assessed camizestrant in combination with a CDK4/6 inhibitor in the first-line metastatic setting. In the phase II SERENA-2 study in 240 previously treated patients, camizestrant demonstrated a statistically significant benefit in the risk of disease progression or death by 42% at a 75 mg dose (with a median PFS of 7.2 vs. 3.7 months) and 33% at a 150 mg dose (with a median PFS of 7.7 vs. 3.7 months) when compared to fulvestrant. For patients with *ESR1* mutations, camizestrant showed an even more meaningful reduction in the risk of disease progression or death with a median PFS of 6.3 vs. 2.2 months at a 75 mg dose (a 45% reduction in a 150 mg dose with a median PFS of 9.2 vs. 2.2 months) compared to fulvestrant (Table 3) [65].

Imlunestrant, another oral SERD, has shown promising efficacy in preclinical studies. The phase I EMBER trial (NCT04188548) evaluated the efficacy of imlunestrant alone, in combination with CDK4/6 inhibitor or in combination with AI for ER+ MBC. The trial indicated that imlunestrant was well tolerated and exhibited encouraging anti-tumor activity without any observed dose-limiting toxicities (DLTs). The ORR was 8%, while the CBR was 42%, and the most observed AEs included nausea, diarrhea, fatigue, and arthralgia [66]. Currently, the phase III EMBER-3 trial (NCT04975308) is evaluating the effectiveness of imlunestrant with or without abemaciclib in comparison to fulvestrant or exemestane, for the treatment of hormone-sensitive advanced breast cancer following progression on aromatase inhibitors. The neoadjuvant setting is also being evaluated in the ongoing window-of-opportunity phase I EMBER-2 trial, examining the pharmacodynamic effect of imlunestrant (NCT04647487) [67].

## 6. Beyond SERD, Clinical Data from CERAN, SERCA and PROTACs

While the focus on developing orally bioavailable SERDs has taken center stage in the development of ER-targeted therapies, additional novel strategies to target ER have also been pursued. OP-1250 is an orally bioavailable CERAN that entered clinical development [68]. It showed positive early monotherapy data in a phase I/II study (NCT04505826) in patients with advanced and/or metastatic HR+, HER2− breast cancer. According to data presented at the 34th EORTC-NCI-AACR Symposium, data from 57 patients evaluated for efficacy indicated that 41% exhibited anti-tumor activity, including a reduction in target lesions. This activity was observed in both wild-type and mutant estrogen receptors, encompassing both the 60 mg and 120 mg doses. Among these patients, six displayed partial responses, with four cases confirmed and two remaining unconfirmed [69].

Olema therapeutics is assessing OP-1250 in a phase II combination therapy trial in combination with the CDK4/6 inhibitor (NCT05266105) and with the PI3Kα inhibitor alpelisib (NCT05508906). Based on the clinical trial data (NCT05266105) presented at the 2023 ESMO Breast Cancer Annual Congress, the combination of OP-1250 and palbociclib produced a tolerable safety profile and elicited tumor responses and disease stabilization in patients with ER+ MBC. Out of the 29 treated patients, one patient achieved a confirmed partial response, and four patients experienced an unconfirmed partial response. Further, nine patients experienced stable disease, including four who had stable disease for at least 24 weeks and the CBR was 41.7% (*n* = 5/12) [69].

The orally bioavailable SERCA, H3B-6545, showed antitumor activity in heavily pretreated patients with ER+ MBC as a single agent. According to a 2021 JCO abstract, patients with *ESR1* Y537S mutations exhibited an ORR of 40% and a median PFS of 7.3 months. Observed adverse events (AEs) included asymptomatic sinus bradycardia and QTc prolongation. It is currently being evaluated in phase I and II settings as a monotherapy (NCT03250676) and in combination with palbociclib (NCT04288089) for ER+ MBC patients who have progressed on prior ET [70].

On the other hand, Arvinas’ Vepdegestrant is the first ER PROTAC to have made its way into clinical development. In preclinical trials, Vepdegestrant exhibited promising results with tumor regression in PDX models with *ESR1* mutations [71]. The latest data from a phase I/II study (NCT04072952) presented at the 2023 ESMO Congress revealed that Vepdegestrant is an effective treatment option for patients with ER+, HER2− advanced breast cancer. Furthermore, the drug exhibited clinical activity across all doses examined, despite the patients being heavily pretreated, with favorable tolerability. At a follow up of 13.8 months, Vepdegestrant showed a CBR of 36.1% across all dose groups (*n* = 83), with a slightly higher CBR (48.8%) observed in patients with *ESR1* mutations (*n* = 43). Out of the 61 patients with measurable disease at baseline, seven experienced a confirmed partial response, resulting in an ORR of 11.5%. The phase III VERITAC-2 trial (NCT05654623) is comparing the safety and efficacy of a once-daily dose of Vepdegestrant to fulvestrant as a second- or third-line treatment [72]. Meanwhile, the phase III VERITAC-3 trial (NCT05909397) is currently assessing the effectiveness of Vepdegestrant paired with palbociclib vs. letrozole plus palbociclib in the first-line treatment for patients with ER+/HER2− advanced breast cancer who have not priorly undergone systemic treatment for their advanced condition. OP-1250, Vepdegestrant, and H3B-6545 have demonstrated promising outcomes in clinical trials, affirming their efficacy. However, comprehensive randomized trials with an approved endocrine therapy comparator are needed to determine whether these alternative mechanisms of ER antagonism offer advantages over the currently authorized agents [73,74].

## 7. Role of CDK4/6 Inhibitor in HR+/HER2− Breast Cancer Treatment

In addition to endocrine therapy, the introduction of CDK4/6 inhibitors has notably reshaped the treatment landscape for advanced HR+ breast cancer. CDK4/6 inhibitors, in combination with ET, remain the standard of care first-line treatment for many patients with HR+/HER2− metastatic breast cancer. Over the past 8 years, these agents have also been widely implemented as adjuvant therapy for patients with higher-risk early-stage disease. Currently, three CDK4/6 inhibitors have received approval from the FDA: palbociclib (Ibrance), ribociclib (Kisqali), and abemaciclib (Verzenio,) (Figure 3). Among these, Palbociclib was the first CDK4/6 inhibitor to demonstrate an improvement in PFS compared to fulvestrant monotherapy in patients who had progressed on prior ET [75]. Both ribociclib and abemaciclib also demonstrated a statistically significant PFS benefit for similar patient populations. This has led to an increase in OS among patients who received CDK4/6i as second-line treatment [76,77]. These drugs are also approved for use in combination with an aromatase inhibitor (AI) as first-line treatment. In fact, all three drugs have shown a significant PFS benefit when compared to AI monotherapy, in this context. The activation of CDK4/6 by cyclin D induces Rb phosphorylation and the progression of the cell cycle into S phase, which is linked with resistance to ET. Hence, CDK4/6 inhibitors have the promise to improve efficacy by blocking the cyclin D-CDK complex, which leads to the prevention of retinoblastoma 1 (Rb1) phosphorylation and ultimately causes G1 phase arrest. When an ET is co-administered with CDK4/6 inhibitors, it results in a synergistic effect that blocks the G1 to S phase cell cycle transition. Hence, the use of ET with a CDK4/6i has become the standard initial therapy for patients with advanced HR+/HER2− breast cancer [78]. Trilaciclib, an intravenously administered CDK4/6 inhibitor, is typically administered prior to chemotherapy. It brings potential safety benefits and antitumor efficacy when combined with cancer chemotherapy, by inducing a temporary and reversible G1 cell cycle arrest in the proliferating hematopoietic stem and progenitor cells in the bone marrow. This helps in shielding them from chemotherapy damage. Early data from a phase II trial (NCT05113966) indicate that Trilaciclib could substantially decrease adverse events when provided before administering sacituzumab govitecan-hziy, also known as Trodelvy, to patients suffering from unresectable, locally advanced, or metastatic triple-negative breast cancer [79].

## 8. Comparison between Clinically Approved CDK4/6 Inhibitors

The design of CDK4/6 inhibitors is intended to exhibit selectivity for CDK4/6 with varying degrees of inhibitory activity against other CDKs. Palbociclib has a comparable affinity for CDK4 and CDK6, with ribociclib and abemaciclib as more potent inhibitors of CDK4. Ribociclib displays the highest relative affinity for CDK4 compared to CDK6. Abemaciclib, on the other hand, differs from ribociclib and palbociclib in its ability to inhibit a broader range of other kinases, including CDK2, although to a lesser degree than CDK4 or CDK6. These varying affinities of the three drugs for CDK4 and CDK6 could contribute to the differences in their toxicity profiles [80,81]. It is believed that the inhibition of CDK4 is responsible for the effectiveness of CDK4/6 inhibitors in treating ER+ breast cancer, as CDK4 is the primary kinase in these cancer cells. On the other hand, CDK6 is the primary kinase in bone marrow progenitors, and inhibiting CDK6 is thought to cause neutropenia. The weak inhibition of CDK2 by abemaciclib could be a crucial factor of why it shows differential activity compared to other agents, like palbociclib and ribociclib. The palbociclib-resistant cells are resistant to ribociclib but still respond to abemaciclib [80,81].

Despite these differences, all of these three inhibitors have demonstrated remarkably consistent median PFS outcomes, and studies like PALOMA, MONALEESA, and MONARCH have shown that this strategy yields superior outcomes compared to ET alone. The expedited approval of the combination of palbociclib and letrozole by the FDA in 2015 was triggered by the findings from PALOMA-1, a randomized phase II trial. This study evaluated palbociclib plus letrozole as a first line treatment and found a statistically significant enhancement in the median PFS [82]. A subsequent phase III PALOMA-2 study further confirmed it. The PALOMA-2 study had a total of 666 postmenopausal participants with HR+, HER2-negative metastatic breast cancer who had received no previous treatment for metastatic disease and were not permitted to have received any prior chemotherapy. These patients were randomly divided into two groups; one group received the CDK4/6 inhibitor palbociclib at a dosage of 125 mg orally once daily for 3 out of 4 weeks, in repeated cycles along with letrozole (2.5 mg, once daily, continuously), while the other group received letrozole and a placebo. The median OS after a median follow-up of 90 months was 53.9 months with palbociclib/letrozole vs. 51.2 months with letrozole alone (hazard ratio [HR] = 0.95). The study previously met its primary endpoint of PFS, which was improved with palbociclib and letrozole to a median of 27.6 months vs. 14.5 months with letrozole alone (HR = 0.580; *p* > 0.001). The study concluded that PALOMA-2 successfully met its primary endpoint of improving PFS, but its secondary endpoint of OS was not statistically significant [83]. The PALOMA-3 trial, involving women with HR+/HER2− advanced breast cancer, demonstrated that regardless of mutation status, palbociclib plus fulvestrant significantly prolonged PFS compared with placebo plus fulvestrant [11.2 vs. 4.6 months, respectively; hazard ratio, 0.50]. With a median follow-up of 44.8 months, palbociclib plus fulvestrant demonstrated a longer OS compared to placebo plus fulvestrant (34.9 vs. 28.0 months, respectively; hazard ratio, 0.81) [84]. In the first-line setting, the results from the phase III MONARCH-3 trial showed that the combination of an aromatase inhibitor (either anastrozole or letrozole) and abemaciclib achieved similar results to those of palbociclib plus letrozole (Table 4). After 8 years of follow up, the MONARCH 3 study demonstrated that women taking abemaciclib and an aromatase inhibitor had an OS exceeding 5.5 years. This represents an increase of 13.1 months compared with the control arm in the intent-to-treat population (66.8 vs. 53.7 months);although, statistical significance for the OS outcome was not reached (hazard ratio [HR] = 0.804. The study maintained a significant median PFS benefit and showed substantial differences in the 6 year PFS rates. Although the statistical significance for the OS outcome was not accomplished, the study achieved PFS statistical significance in an interim analysis in 2017, leading to regulatory approvals for this indication in 2018 [85,86].

The primary aim of the phase III MONALEESA-2 trial was to assess the efficacy of ribociclib plus letrozole in the first-line setting [87]. In a cohort of patients diagnosed with HR+, HER2− advanced breast cancer, characterized by either de novo metastatic disease or late recurrence following neoadjuvant therapy, the frontline combination of ribociclib and letrozole significantly prolonged the OS over letrozole alone. Data presented at the 2023 ESMO Breast Cancer Annual Congress showed a median follow up of 79.8 months, the median OS was 69.2 months, contrasting with 54.3 months for those treated with letrozole alone (*n* = 270). This resulted in a noteworthy 25% relative decrease in the risk of death (HR, 0.75). Additionally, the combination of ribociclib and letrozole exhibited an advancement in PFS compared to letrozole alone, with median PFS figures of 30.3 months and 16.7 months, respectively. This translated to a substantial 43% reduction in the risk of disease progression or death in this population (HR, 0.57), thereby emphasizing the survival advantages associated with ribociclib and letrozole in HR+ breast cancer.

The phase III MONALEESA-7 trial specifically focused on pre-/perimenopausal patients with HR+/HER2− advanced breast cancer and studied ribociclib plus endocrine therapy (ET) vs. placebo plus ET. A notable OS benefit was observed with first line ribociclib + nonsteroidal AI vs. nonsteroidal AI alone (58.7 vs. 40.7 months [HR 0.76]), translating to a 24% relative reduction in the risk of death [88].

In the phase III MONALEESA-3 trial, the efficacy of ribociclib plus fulvestrant was compared with that of fulvestrant on the patients with HR+, HER2− advanced breast cancer and included both first line and second-line patients. The results showed that ribociclib + fulvestrant provided a significant OS benefit over fulvestrant alone (67.6 vs. 51.8 months [HR 0.67], translating to a 33% relative reduction in the risk of death [89].

These three approved CDK4/6 inhibitors also show significant differences in toxicity profiles. In phase I safety evaluations, neutropenia was the primary dose limiting toxicity (DLT) seen with palbociclib and ribociclib, whereas the main DLTs seen with abemaciclib were diarrhea and fatigue [90]. These distinctions in toxicity profiles allow clinicians to tailor the choice of medication to best fit the needs of individual patients.

## 9. Endocrine Therapy Combined with PIK3CA/AKT/mTOR Inhibitors

Recent research indicates a potential link between the PI3K/AKT/mTOR pathway and acquired resistance to endocrine therapy in HR+ breast cancer. Mutations in PIK3CA, which encodes PI3K, have been observed in about 40% of patients with HR+ breast cancer [91]. While the initial treatment is usually a combination of CDK4/6 inhibitor and endocrine therapy, there are other options for subsequent treatment that target the PI3K/AKT/mTOR pathway. Current guidelines propose the utilization of fulvestrant, a SERD, and alpelisib, a PI3K inhibitor, as potential second-line therapies for this patient population [92,93]. The SOLAR-1 trial, a phase III randomized controlled study, investigated the comparison between fulvestrant with alpelisib and fulvestrant with a placebo in HR+ breast cancer. Among patients with PIK3CA mutations, those treated with fulvestrant and alpelisib experienced an extended PFS of 11.0 months compared to 5.7 months with fulvestrant and placebo (hazard ratio, 0.65; *p* < 0.001). The ORR was also higher in the fulvestrant and alpelisib group (26.6% vs. 12.8%). However, this combination did not significantly improve PFS or ORR in patients without PIK3CA mutations [94]. The subsequent BYLieve trial investigated the benefits of alpelisib plus fulvestrant that would be beneficial in patients who had received CDK4/6 inhibitors, which had become a standard therapy after SOLAR-1 completed enrollment. The benefits seen in SOLAR-1 were sustained, with a clinical benefit rate of 50% at 6 months and a median PFS of 7.3 months. These positive outcomes led to the FDA approval of fulvestrant and alpelisib combination therapy for advanced HR+ breast cancer with PIK3CA mutations [95]. However, alpelisib comes with safety concerns, the drug’s main toxicities are hyperglycemia, diarrhea, and rash, for which the rates of grade 3 adverse events were 33% (and 4% grade 4), 7%, and 20%, respectively. For all grades, more than half of patients experienced these toxicities. Elevated blood sugar levels, known as hyperglycemia, result from the desired effect of insulin signaling via the PI3K pathway. Disrupting this pathway may lead to a form of insulin resistance, potentially causing hyperglycemia. While alpelisib, the initial FDA-approved PI3K inhibitor, faces challenges due to its side effects, there is a growing interest in creating alternatives that are more selective and less harmful. Among these alternatives is inavolisib, which operates by binding to ATP, and a new class of allosteric inhibitors designed specifically for PIK3CA mutants, such as LOXO-783 and RLY-2608. Inavolisib is currently in advanced stages of development. These allosteric inhibitors focus on the mutant form of PI3K, minimizing their impact on the wild-type version. This targeted approach aims to enhance tolerability, particularly in addressing hyperglycemia concerns associated with these medications.

The FAKTION trial demonstrated the efficacy of combining capivasertib and fulvestrant to target the AKT pathway. The combination therapy significantly extended both PFS (10.3 vs. 4.8 months; hazard ratio, 0.56; *p* = 0.0023) and OS (29.3 vs. 23.4 months; hazard ratio, 0.66; *p* = 0.035) relative to placebo/fulvestrant in patients with AI-resistant advanced HR+/HER2− breast cancer. These encouraging results prompted the phase III CAPItello-291 trial, which looked into the safety and effectiveness of this joint therapy for patients with AI-resistant advanced HR+/HER2− breast cancer, particularly those who had undergone CDK4/6 inhibitors treatment before. The median PFS with the combined therapy was double that of fulvestrant alone (7.2 months vs. 3.6 months) with a 0.60 hazard ratio, a result statistically significant (*p* < 0.001). Improvement was also seen in patients with AKT alterations, with a median PFS of 7.3 months compared to 3.1 months (hazard ratio = 0.50; *p* < 0.001) [96]. Capivasertib’s primary side effect is diarrhea, observed in about 75% of CAPItello-291 patients, with a grade 3 severity noted in 9%. Nausea ranks as the second most frequent side effect, affecting roughly 35% of individuals. Unlike typical side effects associated with PI3K inhibitors, such as hyperglycemia, capivasertib demonstrates a lower occurrence. Additionally, the prevalence of rashes is minimal. This suggests a potential advantage for AKT inhibitors [97,98].

Everolimus, an mTOR inhibitor, has been studied as a treatment option for advanced HR+ breast cancer patients who have previously been treated with nonsteroidal aromatase inhibitors (AIs). The phase III BOLERO-2 trial yielding a doubling in median PFS in patients with advanced HR+ breast cancer receiving exemestane in combination with everolimus compared to those on exemestane alone (7.8 vs. 3.2 months; hazard ratio, 0.45; *p* < 0.0001) (Table 5). These findings led to an FDA approval of this combination [99,100]. Exemestane was the endocrine partner in BOLERO-2, and outcomes in both arms were much better in patients with wild-type tumors than in those with *ESR1* mutations. There also appeared to be some differences in outcomes based on whether the *ESR1* mutation was D538G or Y5375. Individuals with D538G mutations who underwent treatment with exemestane and everolimus exhibited a PFS similar to that of individuals with a wild-type *ESR1*. These findings underscore the interplay between ER and mTOR signaling and emphasize the value of circulating biomarkers in monitoring therapeutic response and predicting prognosis. Studies indicate that superior outcomes might be attainable through targeted therapy involving a SERD as the endocrine partner. This strategy is currently under investigation in the ELEVATE trial, an umbrella study examining the safety and effectiveness of combining elacestrant with alpelisib, everolimus, palbociclib, abemaciclib, and ribociclib (NCT05563220) [101].

## 10. Conclusions

The year 2023 was a big year for the estrogen receptor (ER), featuring FDA approval of elacestrant and phase III initiation of Vepdegestrant, camizestrant, and OP-1250 as new therapeutic agents. These innovative therapies, with many others, provide a ray of hope for patients, offering an expanded array of options that may enhance the outcomes of treatment. The focus lies on developing highly effective cures that can bypass resistance to current treatments, diminish or delay the requirement for more aggressive chemotherapy, and elevate the quality of life for patients. Therapies of this kind are in high demand and represent a leap forward in the battle against breast cancer. The discovery and advancement of treatments such as oral SERDs, oral CERANs, oral PROTAC, oral SERCA, and many others offer the possibility of more efficient and safer ER-targeted therapies for patients grappling with both early and late-stage ER+ breast cancers.

Over the years, we have gained a detailed understanding of different mutations, their protein location, and the drugs that prompt their emergence. We’ve also discovered how some drugs can cause resistance, or in certain cases, sensitivity to both traditional and newer agents. In the future, there is a need to delve deeper into understanding how these mutations interact with others within the same cell. However, numerous questions still require further exploration. There is a need to identify which of these novel anti-estrogen agents, whether PROTACs or the innovative generation of SERDs, can effectively tackle these *ESR1*, PIK3CA, and AKT mutations. It is also important to determine if there are differences between these mutations, as it is unlikely that they all have equal potential in inducing resistance to treatments or enhancing downstream effects in the ER pathway. The best combinations among the endocrine therapies on the second-line setting needs further exploration. Another crucial yet unresolved question is: Where will these drugs be positioned in the second- and third-line settings? Will there be significant indications for monotherapy effectiveness, or will we be exploring the perfect combination partner? Both strategies probably need further investigation. In this review, we have discussed the next generation of therapeutic agents under evaluation for ER+ breast cancer. We offer our insights regarding the potential opportunities and significant challenges that must be surmounted to unleash this wave of groundbreaking innovation.

## Figures and Tables

**Figure 1 cancers-16-00552-f001:**
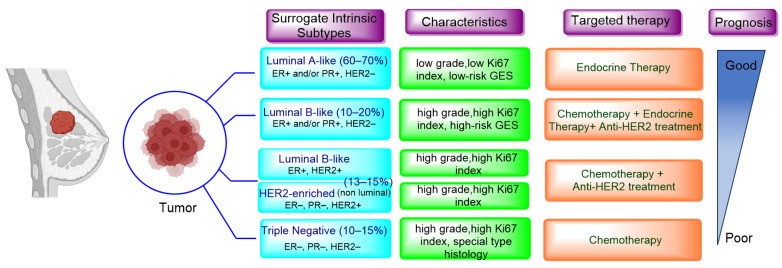
Surrogate intrinsic subtypes, standard of care and prognosis of breast cancer. Adapted from ref [16].

**Figure 2 cancers-16-00552-f002:**
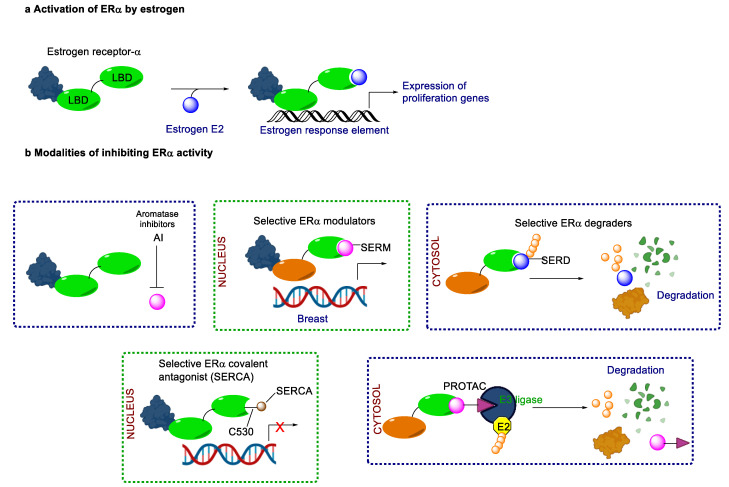
Estrogen receptor-α signaling and modes of inhibition.

**Figure 3 cancers-16-00552-f003:**
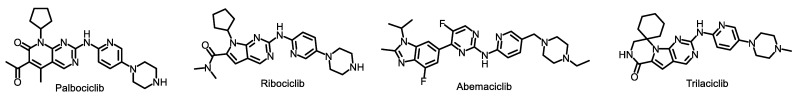
Structures of CDK4/6 inhibitor.

**Table 1 cancers-16-00552-t001:** Structures of new generation Endocrine agents (SERDs, SERCA, CERAN and PROTACs).

ERα Ligand Core	Side Chain	Structures of SERDs, SERCA, CERAN and PROTACs
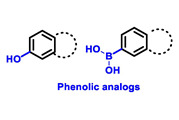	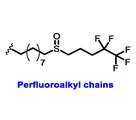	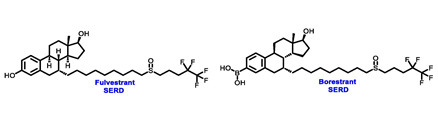
	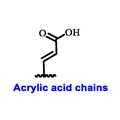	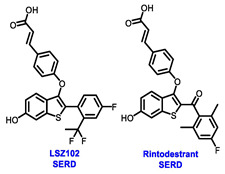
	Secondary or Tertiary amine	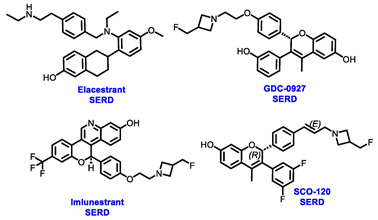
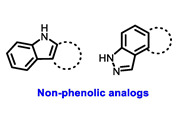	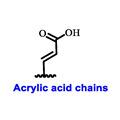	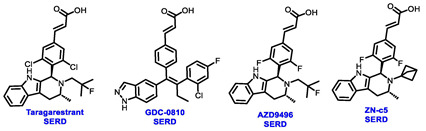
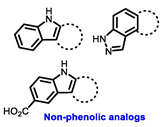	Tertiary amine	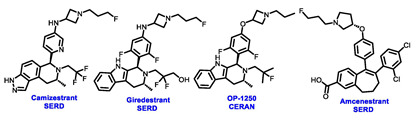
	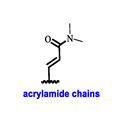	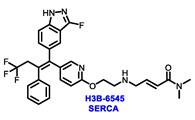
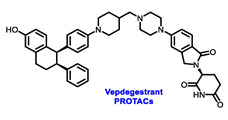		

**Table 2 cancers-16-00552-t002:** Prevalence of *ESR1* mutations in ctDNA and outcomes in metastatic breast cancer.

Trial	Study Treatment	*ESR1* Mutation, %
BOLERO-2 [34]	Exemestane ± everolimus	28.8
FERGI [36]	Fulvestrant ± pictilisib	37.3
PALOMA-3 [37]	Fulvestrant ± palbociclib	25.3
SOFeA [37]	Fulvestrant ± anastrozole	39.1

**Table 3 cancers-16-00552-t003:** SERDs in ER+/HER2− MBC.

Treatment	Elacestrant	Camizestrant	Imlunestrant ± Abemaciclib	Amcenestrant	Giredestrant
Control arm	Fulvestrant/AIs	Fulvestrant	Fulvestrant/exemestane	Fulvestrant/AIs/tamoxifen	Fulvestrant/AIs
Phase (N)	Phase III (477)	Phase II (240)	Phase III (860)	Phase II (290)	Phase II (303)
Patients	Men or postmenopausal women	Postmenopausal women	Men or postmenopausal women	Men or women (any menopausal status)	Men or women (any menopausal status)
Prior CDK4/6i	Required (100%)	Permitted (51%)	Permitted	Permitted (79%)	Permitted (42%)
Allowed prior fulvestrant	Yes	No	No	Yes	Yes
Data readout	Positive (registrational)	Positive (non-registrational)	Ongoing	Negative	Negative

**Table 4 cancers-16-00552-t004:** Selected studies testing CDK4/6 inhibitors in patients with metastatic HR+, HER2−breast cancer.

First-Line Therapy						
Trial	PALOMA-2 (*n* = 666)	PALOMA-3 (*n* = 521)	MONALEESA-2 (*n* = 545)	MONALEESA-3 (*n* = 726)	MONALEESA-7 (*n* = 672)	MONARCH-3 (*n* = 493)
Endocrine partner	Letrozole	Fulvestrant	Letrozole	Fulvestrant	Letrozole, anastrozole, or tamoxifen + LHRH agonist	Letrozole
CDK4/6i	Palbociclib	Palbociclib	Ribociclib	Ribociclib	Ribociclib	Abemaciclib
Study details	AI-naive Patients randomized (2:1) to palbociclib vs. placebo as first-line therapy	AI-pretreated Patients randomized (2:1) to palbociclib vs. placebo as second or later-line therapy	AI-naive Patients randomized (1:1) to ribociclib vs. placebo as first-line therapy	AI-naive and AI-pretreated Patients randomized (2:1) to ribociclib vs. placebo as first-line or second-line therapy	AI-naive and AI-pretreated Patients randomized (2:1) to ribociclib vs. placebo as first-line or second-line therapy	AI-naive Patients randomized (2:1) to abemaciclib vs. placebo as first-line therapy
Median PFS, CDK4/6i + ET vs. ET, (month)	27.6 vs. 14.5 (Δ13.1)	11.2 vs. 4.6 (Δ6.6)	30.3 vs. 16.7 (Δ13.6)	20.5 vs. 12.8 (Δ7.7)	23.8 vs. 13.0 (Δ10.8)	29.0 vs. 14.8 (Δ14.2)
Hazard ratio	0.56	0.50	0.57	0.59	0.55	0.54
Median OS, CDK4/6i + ET vs. ET (month)	53.9 vs. 51.2	34.9 vs. 28.0	69.2 vs. 54.3	67.6 vs. 51.8	58.7 vs. 48.0	66.8 vs. 53.7 (interim analysis)
Hazard ratio	0.95	0.81	0.75, Significant	0.67, Significant	0.76, Significant	0.80

**Table 5 cancers-16-00552-t005:** Phase II/III trials of PI3K/AKT/mTOR inhibitors in advanced breast cancer.

Study (Phase)	Population	Arms	mPFS (Months)
SOLAR-1 (III)	HR+/HER2−	Alpelisib + FLV vs. placebo + FLV	PIK3CA mut: mPFS 11.0 alpelisib vs. 5.7 placebo, *p* < 0.001 PIK3CA not mut: mPFS 7.4 alpelisb vs. 5.6 placebo HR 0.85
FAKTION (II)	HR+/HER2−	Capivasertib + FLV vs. FLV + placebo	mPFS 10.3 capivasertib vs. 4.8 placebo, *p* = 0.0018
BOLERO-2 (III)	HR+/HER2−	Exemestane + everolimus vs. Exemestane + placebo	mPFS 6.9 everolimus vs. 2.8 placebo, *p* < 0.001

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
