# Peer review of "Therapies for the Treatment of Advanced/Metastatic Estrogen Receptor-Positive Breast Cancer: Current Situation and Future Directions"

_cancers, 2024, doi:10.3390/cancers16030552_

Round 1

Reviewer 1 Report

Comments and Suggestions for Authors

This is a review article by Rej at al summarising next generation therapies targeting the ER in breast cancer, which is a very interesting and fast-moving topic. However, I feel that the manuscript in its present form is quite lengthy and a bit confusing, I think it would benefit from streamlining exactly what the authors wants to describe. The title suggests that the review will be an overview of ER therapies in breast cancer in general. However, the content is largely related to ER therapies in metastatic/ advanced breast cancer only (rather than primary/neoadjuvant/adjuvant settings) and there is a lot of detail about TNBC  and HER2+ disease. I believe this makes the direction of the article unclear and therefore the conclusions difficult. The article would benefit from reshaping and either focusing on 1. ER therapies in metastatic/advanced breast cancer in ER+ disease 2. ER therapies in ER+ breast cancer as a whole. The subject of relevant therapies in TNBC or HER2+ disease is the content of a different article altogether. 

I have made some more specific comments about the article below, but overall I think it needs some significant changes to the layout and content before going in to the finer details. 

Abstract

·         Line 30/31 I find this statement a bit confusion. Advancements in the treatment of breast cancer have led to a reduction in mortality – this is correct. I am not sure what the next part means ‘ an overall decline of 43%’ – are you saying mortality has declined by 43% if so from what to what% - the figure does not mean much on its own

·         I appreciate that it is not possible to outline all of the potential treatments in the abstract but actual therapies are not mentioned at all – please include a line summarising the therapies which will be discussed – e.g. CDK 4/6 inhibitors etc

Introduction

·         Lines 44 – 46 could we have a reference please as to where the percentage figures have come from – also it is standard practice to use ‘%’ in the text instead of ‘percent’

·         Lines 46 – 51 can be summarised, the authors have defined luminal A subtype twice which is not necessary

·         Lines 53 – in contrast to luminal A, there is no definition of luminal B. The whole initial paragraph needs rewording to briefly explain the subtypes – this should be well known to the reader so does not need too much detail

·         Lines 58 – 85 a lot of time is spent discussing TNBC and HER2 positive disease – however the aim of the article is ER+ disease

·         The whole introduction needs to be reworded to emphasise the key points in ER+ disease and briefly summarise the important points in TNBC and HER2+ only and then explaining that the rest of the article is going to go into

·         Figure 1: I think there is some confusion here and in the introduction regarding primary and adjuvant therapy. The legend and diagram are suggesting that endocrine therapy is the first line of treatment for luminal A subtype which is not correct – it would be surgery followed by appropriate adjuvant endocrine therapy

Endocrine therapy – treatment for ER+/HER2- MBC

·         Some of this information could be moved into the introduction e.g. the four classes of drugs you are going to talk about. Then in this section you should talk about the four classes of drugs in relation to ER+/HER2- disease (rather than mechanism of action etc)

Prevalence of ESR1 mutations in metastatic breast cancer

·         Why the focus on just metastatic breast cancer? Both this section and the above focus solely on metastatic disease, but then in the rest of the article the authors appears to be talking about ER therapies in general, this is confusing.

The title and abstract of the article suggests that this is going to be a discussion about ER therapies in general for breast cancer, however, the main focus of the article is on metastatic breast cancer. There is also discussion on advanced breast cancer without clearly defining what is considered advanced versus metastatic and how this can potentially impact treatment.

Section 9 CDK4/6 inhibitors in TNBC I am not sure is entirely relevant if the focus of the article is on ER+ disease? Particularly as the article is quite lengthy in its current form. Also section 11, again discusses TNBC only

Comments on the Quality of English Language

No concerns

Author Response

Reviewer 1 comments: This is a review article by Rej et al summarising next generation therapies targeting the ER in breast cancer, which is a very interesting and fast-moving topic. However, I feel that the manuscript in its present form is quite lengthy and a bit confusing, I think it would benefit from streamlining exactly what the authors wants to describe. The title suggests that the review will be an overview of ER therapies in breast cancer in general. However, the content is largely related to ER therapies in metastatic/ advanced breast cancer only (rather than primary/neoadjuvant/adjuvant settings) and there is a lot of detail about TNBC and HER2+ disease. I believe this makes the direction of the article unclear and therefore the conclusions difficult. The article would benefit from reshaping and either focusing on 1. ER therapies in metastatic/advanced breast cancer in ER+ disease 2. ER therapies in ER+ breast cancer as a whole. The subject of relevant therapies in TNBC or HER2+ disease is the content of a different article altogether.
We appreciate the reviewer’s time reading our manuscript and providing constructive comments. A point-by-point response to the comments made by the reviewers is shown below.
Based on the reviewer’s suggestion the review article has been reshaped to focus on ER therapies in metastatic/advanced breast cancer in ER+ disease. The section describing relevant therapies in TNBC or HER2+ has now been removed. Accordingly, the title of the review has also been changed to “Therapies for the Treatment of Advanced /Metastatic Estrogen Receptor-Positive Breast Cancer: Current situation and Future Directions”.
Q1. Line 30/31 I find this statement a bit confusion. Advancements in the treatment of breast cancer have led to a reduction in mortality this is correct. I am not sure what the next part means ‘an overall decline of 43%’ – are you saying mortality has declined by 43% if so from what to what% - the figure does not mean much on its own.
A1. This line has been replaced by “Advancements in the treatment breast cancer, have decreased death rates by 43% from 1989 to 2020, translating to 460,000 fewer breast cancer deaths during that time.” Also, this line was removed from the abstract to the introduction section.
Q2. I appreciate that it is not possible to outline all of the potential treatments in the abstract but actual therapies are not mentioned at all – please include a line summarising the therapies which will be discussed – e.g. CDK 4/6 inhibitors etc
A2. Thank you for the suggestion. The abstract has been rewritten as follows.
New abstract- “The hormone receptor-positive (HR+) type is the most frequently identified subtype of breast cancer. HR+ breast cancer has a more positive prognosis when compared to other subtypes, such as human epidermal growth factor protein 2-positive disorder and triple-negative disease. The advancement in treatment outcomes for advanced HR+ breast cancer has been considerably elevated due to the discovery of cyclin-dependent kinase 4/6 inhibitors and their combination effects with endocrine therapy. However, despite the considerable effectiveness of tamoxifen, a selective estrogen receptor modulator (SERMs), and aromatase inhibitors (AI), the issue of treatment resistance still presents a significant challenge for HR+ breast cancer. As a result, there is a focus on exploring new therapeutic strategies such as targeted protein degradation and covalent inhibition for targeting ERα. This article discusses the latest progress in treatments like oral selective ER degraders (SERDs), complete estrogen receptor antagonists (CERANs), selective estrogen receptor covalent antagonists (SERCAs), proteolysis targeting chimera (PROTAC) degraders, and combinations of CDK4/6 inhibitors with endocrine therapy. The focus is specifically on those compounds that have transitioned into phases of clinical development.”
Q3. Lines 44 – 46 could we have a reference please as to where the percentage figures have come from – also it is standard practice to use ‘%’ in the text instead of ‘percent’.
A3. We have replaced ‘percent’ by % and required reference has been included as “Luminal A-like accounts for 60-70 % of breast cancer cases in the United States, followed by Luminal B-like HER2- at 10-20 %, triple negative at 10-15 %, and HER2-enriched and Luminal B-like HER2+ together at approximately 13-15 % (Figure 1).[1,2]”
Q4. Lines 46 – 51 can be summarised, the authors have defined luminal A subtype twice which is not necessary.
A4. Repetitive luminal A subtype description has been removed from the introduction section.
Q5. Lines 53 – in contrast to luminal A, there is no definition of luminal B. The whole initial paragraph needs rewording to briefly explain the subtypes – this should be well known to the reader so does not need too much detail.
A5. We have reworded the initial paragraph, and the necessary description of luminal B has been included.
Q6. Lines 58 – 85 a lot of time is spent discussing TNBC and HER2 positive disease – however the aim of the article is ER+ disease. The whole introduction needs to be reworded to emphasise the key points in ER+ disease and briefly summarise the important points in TNBC and HER2+ only and then explaining that the rest of the article is going to go into.
A6. The review article has been reshaped to focus on ER therapies in metastatic/advanced breast cancer in ER+ disease, and the section describing detailed therapies for TNBC and HER2+ has been removed. Summarized important points in TNBC and HER2+ has been included in the introduction.

Q7. Figure 1: I think there is some confusion here and in the introduction regarding primary and adjuvant therapy. The legend and diagram are suggesting that endocrine therapy is the first line of treatment for luminal A subtype which is not correct – it would be surgery followed by appropriate adjuvant endocrine therapy.
A7. We have included a new Figure 1.
Q8. Endocrine therapy – treatment for ER+/HER2- MBC.
Some of this information could be moved into the introduction e.g. the four classes of drugs you are going to talk about. Then in this section you should talk about the four classes of drugs in relation to ER+/HER2- disease (rather than mechanism of action etc).
A8. Thank you for the suggestion. The section has been reworded.
Q9. Prevalence of ESR1 mutations in metastatic breast cancer.
Why the focus on just metastatic breast cancer? Both this section and the above focus solely on metastatic disease, but then in the rest of the article the authors appear to be talking about ER therapies in general, this is confusing.
Q10. The title and abstract of the article suggests that this is going to be a discussion about ER therapies in general for breast cancer, however, the main focus of the article is on metastatic breast cancer. There is also discussion on advanced breast cancer without clearly defining what is considered advanced versus metastatic and how this can potentially impact treatment.
Q11. Section 9 CDK4/6 inhibitors in TNBC I am not sure is entirely relevant if the focus of the article is on ER+ disease? Particularly as the article is quite lengthy in its current form. Also, section 11, again discusses TNBC only.
A9-11. The review article has been reshaped, the title has also been changed and in updated form, it focuses on treatment modalities of advanced/metastatic ER+ breast cancer. Previously mentioned sections 9 and section 11 have been removed.

Reviewer 2 Report

Comments and Suggestions for Authors

The title of the submitted manuscript indicates an interesting subject. However the title covers only about two thirds of the manuscript. The authors are discussing all options for ER positive BC in length and then, like an appendix. also include thoughts regarding TNBC, but much more superficial than the parts about ER positive BC. The frame of this paper should be critically reviewed. In the current form it cannot be accepted.

Abstract:

The statement in the abstract (last sentence) sounded as if the the authors wanted to summarize the development of ALL therapeutic modalities for breast cancer. In fact the subject of the review is endocrine and endocrine-based therapy. The abstract should focus on that. Furthermore the abstract should summarize the content of the article. The submitted abstract is rather a pre-introduction that an abstract. It needs to be completely rewritten.

1. Introduction:

The molecular subtypes can only be diagnosed by PAM50, the IHC surrogates the authors reference to are widely used to describe these subtypes but this is scientifically not correct. The authors should clearly state that immunohistochemistry is only a surrogate and can only result in subtypes like "luminal-A-like". In the manuscript it sounds as if the IHC is defining the molecular subtype.

Dual blockade is not only used in the neoadjuvant setting, it is also approved in the adjuvant setting.

The subject of the introduction is TNBC and HER2 positive breast cancer while the subject of the paper is targeting the ER. What is the association between the introduction and the rest of the manuscript? It should be completely rewritten or deleted.

2. Endocrine therapy:

CDK4/6i are not a category of endocrine therapies, in fact they are no ET at all but combination partners of endocrine agents in endocrine-based regimens. CDK4/6i are also working in TNBC (i.e. Trilaciclib). All other principles described in this paragraph are ET and in the figure the CDK4/6i are not included. My recommendation is to remove CDK4/6i from this part. Yours sections 7 following are discussing combination partners of ET in length, it is enough to discuss it there.

9. CDK4/6i and TNBC

Please also discuss the data regarding trilaciclib in this context.

11. TNBC

This section has nothing to do with the subject of this review. If you want to include this you have to change the title and discuss the options as broad as you did with new endocrine therapies. 

Figure 5: This algorithm has nothing to do with the submitted review and it is also not reflecting current therapy options (e.g. ICI for the neoadjuvant therapy of TNBC). I would recommend to completely remove it.

In general the parts about endocrine agents and combinations are well written and are covering the subject. Parts about HER2 positive BC and TNBC should be removed. Then the title can stay as it is. The abstract harbors little information and is more like a pre-introduction. The introduction itself has nothing to do with the rest of the review. At no point the authors explain why they wrote this review and they want to achieve. Unfortunately asks him/herself the same questions. This urgently needs to be addressed and for me is a major flaw.

Author Response

Reviewer 2 comments: The title of the submitted manuscript indicates an interesting subject. However the title covers only about two thirds of the manuscript. The authors are discussing all options for ER positive BC in length and then, like an appendix. also include thoughts regarding TNBC, but much more superficial than the parts about ER positive BC. The frame of this paper should be critically reviewed. In the current form it cannot be accepted.
We appreciate the reviewer’s time reading our manuscript and providing constructive comments. Point-by-point response to the comments made by the reviewer is shown below.
In agreement to the reviewer’s suggestion the review article has been reshaped to focus on ER therapies in metastatic/advanced breast cancer in ER+ disease. The section describing relevant therapies in TNBC or HER2+ has now been removed. Accordingly, the title of the review has also been changed to “Therapies for the Treatment of Advanced /Metastatic Estrogen Receptor-Positive Breast Cancer: Current situation and Future Directions”.
Q1. Abstract: The statement in the abstract (last sentence) sounded as if the authors wanted to summarize the development of ALL therapeutic modalities for breast cancer. In fact the subject of the review is endocrine and endocrine-based therapy. The abstract should focus on that. Furthermore the abstract should summarize the content of the article. The submitted abstract is rather a pre-introduction that an abstract. It needs to be completely rewritten.
A1. We have rewritten the abstract. The new abstract is,
“The hormone receptor-positive (HR+) type is the most frequently identified subtype of breast cancer. HR+ breast cancer has a more positive prognosis when compared to other subtypes, such as human epidermal growth factor protein 2-positive disorder and triple-negative disease. The advancement in treatment outcomes for advanced HR+ breast cancer has been considerably elevated due to the discovery of cyclin-dependent kinase 4/6 inhibitors and their combination effects with endocrine therapy. However, despite the considerable effectiveness of tamoxifen, a selective estrogen receptor modulator (SERMs), and aromatase inhibitors (AI), the issue of treatment resistance still presents a significant challenge for HR+ breast cancer. As a result, there is a focus on exploring new therapeutic strategies such as targeted protein degradation and covalent inhibition for targeting ERα. This article discusses the latest progress in treatments like oral selective ER degraders (SERDs), complete estrogen receptor antagonists (CERANs), selective estrogen receptor covalent antagonists (SERCAs), proteolysis targeting chimera (PROTAC) degraders, and combinations of CDK4/6 inhibitors with endocrine therapy. The focus is specifically on those compounds that have transitioned into phases of clinical development.”

Q2. Introduction: The molecular subtypes can only be diagnosed by PAM50, the IHC surrogates the authors reference to are widely used to describe these subtypes but this is scientifically not correct. The authors should clearly state that immunohistochemistry is only a surrogate and can only result in subtypes like "luminal-A-like". In the manuscript it sounds as if the IHC is defining the molecular subtype. Dual blockade is not only used in the neoadjuvant setting, it is also approved in the adjuvant setting. The subject of the introduction is TNBC and HER2 positive breast cancer while the subject of the paper is targeting the ER. What is the association between the introduction and the rest of the manuscript? It should be completely rewritten or deleted.
A2. Thank you for the suggestion. The introduction has been reshaped to focus on ER therapies in metastatic/advanced breast cancer in ER+ disease, and the section describing detailed therapies for TNBC and HER2+ has been removed.
Q3. Endocrine therapy: CDK4/6i are not a category of endocrine therapies, in fact they are no ET at all but combination partners of endocrine agents in endocrine-based regimens. CDK4/6i are also working in TNBC (i.e. Trilaciclib). All other principles described in this paragraph are ET and in the figure the CDK4/6i are not included. My recommendation is to remove CDK4/6i from this part. Yours sections 7 following are discussing combination partners of ET in length, it is enough to discuss it there.
A3. Thank you for the suggestion. We have made the necessary changes as recommended.
Q4. CDK4/6i and TNBC Please also discuss the data regarding trilaciclib in this context.
A4. Previous section 9 has been removed. Discussion regarding Trilaciclib has been added in new section 7, “Role of CDK4/6 Inhibitor in HR+/HER2- Breast Cancer Treatment.”
Q5. TNBC This section has nothing to do with the subject of this review. If you want to include this you have to change the title and discuss the options as broad as you did with new endocrine therapies.
A5. As recommended section 11 has been removed.
Q6. Figure 5: This algorithm has nothing to do with the submitted review and it is also not reflecting current therapy options (e.g. ICI for the neoadjuvant therapy of TNBC). I would recommend to completely remove it.
A6. As recommended figure 5 has been removed.

Q7. In general the parts about endocrine agents and combinations are well written and are covering the subject. Parts about HER2 positive BC and TNBC should be removed. Then the title can stay as it is. The abstract harbors little information and is more like a pre-introduction. The introduction itself has nothing to do with the rest of the review. At no point the authors explain why they wrote this review and they want to achieve. Unfortunately asks him/herself the same questions. This urgently needs to be addressed and for me is a major flaw.
A7. Thank you for the comment. The review has been reshaped to focus on ER therapies in metastatic/advanced breast cancer in ER+ disease, and the section describing detailed therapies for TNBC and HER2+ has been removed. We have also added our perspective on this topic in the conclusion section.

Round 2

Reviewer 1 Report

Comments and Suggestions for Authors

Authors have responded to comments appropriately

Reviewer 2 Report

Comments and Suggestions for Authors

All comments have been addressed accordingly. I have no further issues.